# Modulation of Mitochondrial Metabolic Reprogramming and Oxidative Stress to Overcome Chemoresistance in Cancer

**DOI:** 10.3390/biom10010135

**Published:** 2020-01-14

**Authors:** Rosario Avolio, Danilo Swann Matassa, Daniela Criscuolo, Matteo Landriscina, Franca Esposito

**Affiliations:** 1CRG-Centre for Genomic Regulation, 08003 Barcelona, Spain; 2Department of Molecular Medicine and Medical Biotechnology, University of Naples Federico II, 80131 Naples, Italy; 3Medical Oncology Unit, Department of Medical and Surgical Sciences, University of Foggia, 7100 Foggia, Italy; 4Laboratory of Pre-Clinical and Translational Research, IRCCS, Referral Cancer Center of Basilicata, 85028 Rionero in Vulture, Italy

**Keywords:** cancer metabolic reprogramming, oxidative stress, drug resistance, tumor necrosis factor receptor associated protein 1 (TRAP1), heat shock protein 90 (HSP90), targeting metabolism for cancer therapy

## Abstract

Metabolic reprogramming, carried out by cancer cells to rapidly adapt to stress such as hypoxia and limited nutrient conditions, is an emerging concepts in tumor biology, and is now recognized as one of the hallmarks of cancer. In contrast with conventional views, based on the classical Warburg effect, these metabolic alterations require fully functional mitochondria and finely-tuned regulations of their activity. In turn, the reciprocal regulation of the metabolic adaptations of cancer cells and the microenvironment critically influence disease progression and response to therapy. This is also realized through the function of specific stress-adaptive proteins, which are able to relieve oxidative stress, inhibit apoptosis, and facilitate the switch between metabolic pathways. Among these, the molecular chaperone tumor necrosis factor receptor associated protein 1 (TRAP1), the most abundant heat shock protein 90 (HSP90) family member in mitochondria, is particularly relevant because of its role as an oncogene or a tumor suppressor, depending on the metabolic features of the specific tumor. This review highlights the interplay between metabolic reprogramming and cancer progression, and the role of mitochondrial activity and oxidative stress in this setting, examining the possibility of targeting pathways of energy metabolism as a therapeutic strategy to overcome drug resistance, with particular emphasis on natural compounds and inhibitors of mitochondrial HSP90s.

## 1. Introduction

Drug resistance is the major cause of cancer recurrence and metastasis, and involves different molecular mechanisms/targets affecting the events that are essential to ensure cell survival. Dissecting the complexity of this process is crucial, both for the development of new effective drugs, and to find the right therapeutic-drug combination to kill cancer cells [1]. In this view, many advances have been made to identify the so-called hallmarks of cancer, an intricate network of mechanisms which are responsible for tumor development and growth [2]. Among those, more and more light has been shed on metabolic reprogramming, a set of mechanisms used by cancer cells to modify their metabolism and adapt it to increased growth requirements, thus providing them with an overall growth advantage compared to their normal cell counterparts. Indeed, it has been shown only recently that several tumors rely on mitochondrial respiration rather than glycolysis, as previously thought, according to the so-called Warburg effect [3,4]. Moreover, it has emerged that oxidative phosphorylation (OXPHOS) is essential for the survival and proliferation of chemoresistant cells [5]. Cell metabolism and energy production are regulated by mitochondria in almost all eukaryotic cells. Mitochondria arises from the endosymbiotic relationship between aerobic bacteria and primordial nucleus-containing host cells [6]. These ancient organelles show different shapes and sizes according to cell type, through a constant balance between mechanisms of fission and fusion [7]. Additionally, the number and volume occupied by mitochondria is quite variable; it mostly depends on the bioenergetic demands of a cell [8]. As a consequence of the endosymbiotic process of prokaryotic cell internalization, mitochondria show a double membrane system, i.e., an outer membrane and an inner membrane, separated by an intramembrane space. While the outer membrane is quite permeable, allowing the diffusion of ions and molecules, the inner membrane is permeable only to small, uncharged molecules. The outer membrane surface is enriched in voltage-dependent anion channels (VDACs), as well as in the proteins necessary for the import of nuclear-encoded proteins. The inner membrane has a larger surface compared to the outer membrane, and contains many finger-like projections protruding into the matrix called cristae. These structures host all the respiratory chain components; therefore, their number reflects the respiratory activity of a cell. This double membrane system surrounds the mitochondrial matrix, a space enriched in proteins involved in substrate metabolism, including those that are essential for fatty acid oxidation and the citric acid cycle [9]. The matrix also contains multiple copies of mitochondrial DNA (mtDNA), a ~16.6 kilobases genome containing 37 genes encoding 13 polypeptides, 2 ribosomal RNAs, and 22 tRNAs [10]. The vast majority of mitochondria-localized proteins are encoded by the nuclear genome through cotranslational, protein-import mechanisms, highlighting the tight regulation of mitochondrial respiration [11]. In addition to ATP generation, which is essential for most of the energy-consuming processes within the cell, mitochondria are necessary for several other functions, including the degradation of biomolecules, as in fatty acid β-oxidation and the Krebs cycle, as well as the regulation of Ca^2+^ homeostasis, innate immunity, and apoptosis [9]. Finally, mitochondria are a major source of reactive oxygen species (ROS), as they generate almost 90% of the total cellular ROS, whose increase leads to oxidative stress and related complications [12]. Therefore, is not surprising that mitochondrial dysfunction contributes to several diseases, including neurodegenerative and metabolic disorders, as well as cancer chemoresistance [13,14]. Although very little is known about the mechanisms of mitochondrial adaptation to such phenomena, many studies have suggested that, among others, mitochondrial fusion and fission processes, known as mitochondrial dynamics, play fundamental roles in the development of chemoresistance [15]. Accordingly, the inhibition of BCL-2/BCL2L1 by the small molecule drug ABT-737 enhances sensitivity to cisplatin by triggering mitochondrial fission and mitophagy in cisplatin-insensitive cholangiocarcinoma cells and in metastatic melanoma cells [15]. Moreover, it has been shown that ROS-mediated mitochondrial fission leads to increased survival of hepatocellular carcinoma (HCC) cells, and that the downregulation of Six1 by miR-488 inhibits Drp1-mediated mitochondrial fission, resulting in the sensitization of ovarian cancer cells to cisplatin [15]. An additional mechanism responsible for cancer chemoresistance is the retrograde (mitochondria-to-nucleus) signaling pathway, activated by mitochondrial dysfunction, that modifies nuclear gene expression and, ultimately, cell behavior. These pathways are usually driven either by ROS or Ca^2+^ levels as a consequence of mtDNA mutations, that, in turn, regulate Akt-, mitogen activated kinase-like protein (MAPK)-, and hypoxia inducible factor 1 subunit alpha (HIF-1α)-mediated signaling, often linked to resistance to drug treatments [13].

In addition to the aforementioned ones, the main mitochondrial mechanisms responsible for cancer chemoresistance are those related to the regulation of cancer cell metabolism. In fact, different cancer cell types may undergo different bioenergetic changes, some toward more glycolytic conditions, and some to more oxidative, depending on their needs in terms of overcoming stress conditions. In this context, increasing evidence shows that, among others, the mitochondrial chaperone TRAP1 (tumor necrosis factor-associated protein 1) seems to be one of the key players, through its ability to promote a stress-adaptive response in cancer cells during oxidative stress and metabolic reprogramming [3]. In this review, we will highlight the biological functions of mitochondria in health and disease, with a particular focus on the mitochondrial mechanisms contributing to cancer chemoresistance.

## 2. The Role of Mitochondria in ATP Synthesis and Reactive Oxygen Species Production

Each fundamental activity of the cell requires energy, from movement to cell division and protein synthesis. Mitochondria represent the main energy source in the cell. This energy comes from the ATP molecules produced by OXPHOS. The OXPHOS system comprises five enzymes, assembled in huge multisubunit proteins, known as complex I (NADH:ubiquinone oxidoreductase), complex II (succinate:ubiquinone oxidoreductase), complex III (ubiquinone:cytochrome c oxidoreductase), complex IV (cytochrome c oxidase), and complex V (ATP synthase). Except for complex II, whose subunits are all encoded by the nuclear genome, all the other complexes are formed by subunits encoded by both the nuclear and the mitochondrial genome [16]. Altogether, the respiratory complexes form the electron transport chain (ETC), that creates an electrochemical proton gradient from the matrix across the mitochondrial inner membrane, resulting in the generation of the mitochondrial membrane potential that is used by the ATP synthase (complex V) to produce ATP from ADP and inorganic phosphate [16]. The only complex that does not take part in the proton pump is complex II. Instead, it serves as an entry point for electrons into the respiratory chain and as a site for the regulation of the activity of the tricarboxylic acid (TCA) cycle [17]. All the other metabolic processes that occur within mitochondria, like fatty acid β-oxidation and the TCA cycle, supply mitochondria with substrates that help to maintain the reduced state of the NADH/NAD^+^ and FADH_2_/FAD couples along the OXPHOS. Since the redox potentials of electron donors and acceptors gradually increase, free energy is released at each step along the chain, driving conformational changes of the complexes (especially I and IV) that facilitate the unidirectional flow of the protons across the inner membrane [17,18]. During OXPHOS, oxygen molecules are partially reduced, leading to the formation of ROS [19]. This term refers to all molecular oxygen-derived free radicals like superoxide anion radicals (•O2−) and hydroxyl radicals (HO•), as well as nonradical species such as hydrogen peroxide (H_2_O_2_). In principle, the reduction of oxygen molecules leads to the formation of •O_2_^−^, which can be then converted into other reactive species such as HO• and H_2_O_2_ [9]. Complexes of the ETC produce the vast majority of ROS, especially complex I and III [20]. In addition, many other mitochondrial enzymes contribute to ROS generation like pyruvate dehydrogenase complex, α-ketoglutarate dehydrogenase complex, monoamine oxidase, glycerol-3-phosphate dehydrogenase, and dihydroorotate dehydrogenase [21]. The level of ROS needs to be constantly balanced to avoid damage to proteins, mtDNA, and lipids, and related consequences. Indeed, mitochondria have two major antioxidant systems including mitochondrial superoxide dismutase (SOD2) and the thioredoxin system, made of thioredoxin-1 (Trx1), thioredoxin-2 (Trx2), thioredoxin reductase-2 (TrxR2), and peroxidase 3 (Prx3) [22].

As mitochondria are ROS producers and, at the same time, can release proapoptotic factors, thiol redox regulation by these systems is strictly involved in cancer cell death. Accordingly, the upregulation of Trx1, Trx2, TrxR2, and Prx3 is considered a negative prognostic factor in several types of cancers, and is observed in drug-resistant tumors, favoring cell growth and preventing apoptosis [23]. This is not surprising, as cancer cells often display high levels of ROS. In particular, it was shown that TrxR2 has a peculiar role in hepatocarcinoma, prostate cancer, osteosarcoma, colon and rectal cancer, as well as in the progression of lung cancer, as it can influence cell proliferation, invasion, and migration [23]. In line with this evidence, Li and coworkers demonstrated that the overexpression of Trx1 in diffuse large B cell lymphoma cell lines is associated with growth and survival [24]. Moreover, high basal expression of Trx1 in tumor cells or in the stromal cells of the surrounding microenvironment may lead to the upregulation of critical drug resistance genes and the development of chemoresistance. Thus, targeting Trx1 with the novel Trx1 inhibitor PX-12 may reverse chemoresistance [24]. Similarly, Prx3 is significantly upregulated in most common malignancies including breast cancer, hepatocellular carcinoma, malignant mesothelioma, lung cancer, cervical cancer, colorectal neoplasm, prostate cancer, and endometrial cancer, and exerts functions in ROS removal, apoptosis, and chemoresistance [25].

Even though ROS have been mainly associated with oxidative stress and damage, increasing evidence showed that these molecules also play a biological role in several physiological functions, such as adaptation to hypoxia, autophagy, immunity, differentiation, and aging [26].

## 3. The Antioxidant Response in Cancer

Many tumors upregulate protective antioxidant pathways in response to elevated ROS levels. For example, oncogenic K-Ras, B-Raf, and c-Myc actively inhibit ROS generation through regulation of nuclear factor (erythroid-derived 2)-like 2 (NRF2), which is considered one of the main orchestrators of the cellular antioxidant response. By controlling the basal and inducible expression of over 200 genes that contain antioxidant response elements, NRF2 plays a variety of roles in cancer development and progression, from proliferation, apoptosis, and angiogenesis, to immunity and inflammation, genome instability, and metabolic reprogramming [27]. Among the genes regulated by NRF2, several control the synthesis of the most abundant cellular antioxidant defense, the tripeptide glutathione (GSH), a quite abundant scavenger whose expression falls within a range between 1–10 mM in all tissue. In particular, in physiological conditions, the levels of the reduced form of GSH are the highest, whereas in the presence of a high level of H_2_O_2_ and lipid peroxide, reduced GSH is oxidized to GSSG, a reaction catalyzed by the GSH peroxidase. GSSG, in turn, can be reduced back to GSH in presence of NADPH. Therefore, the ratio between reduced form glutathione and the oxidized one (GSH/GSSG) is a clear indicator of the cellular redox state. The GSH antioxidant effect is essential in the regulation of several processes, such as cell proliferation, inflammation, apoptosis, as well as the detoxification of xenobiotics [28]. Accordingly, the role of GSH, both as a biomarker and as a therapeutic target in cancer, has been deeply investigated. In general, high GSH levels in tumors are considered a negative prognostic indicator, and accordingly, tumors with the highest GSH display worse prognoses. In fact, the ability of GSH to remove carcinogens can confer resistance to several drugs within tumors with high levels of the tripeptide; moreover, elevated GSH levels can promote cancer cell proliferation, tumor progression, and metastasis by reducing excessive ROS accumulation which, in contrast, can exert beneficial effects in cancer cells by promoting cell death through the damage of several biomolecules. Therefore, cancer cells need to maintain an intricate balance of antioxidant levels to survive [28]. In line with the impact that cell metabolism has on ROS production and the antioxidant response, the NRF2 rewiring of gene expression also comprises the regulation of the enzymes involved in the pentose phosphate pathway (PPP) and glutamine metabolism, both of which are essential for the production of GSH [29]. The PPP is an alternative pathway to glycolysis which cells rely on to synthetize nucleotides. This pathway consists of two phases, the oxidative and the nonoxidative, both of which are connected to glycolysis at the level of several phosphate intermediates. The oxidative phase (which is nonreversible) leads to the production of ribulose-5-phosphate that is converted in ribose-5-phosphate during the nonoxidative phase (consisting of a set of reversible reactions). Therefore, ribose-5-phosphate can finally be used for nucleotide synthesis. In particular, the first reaction of the oxidative PPP, catalyzed by the glucose-6-phosphate dehydrogenase, produces NADPH molecules which are necessary for the regeneration of GSH [30]. In this regard, PPP plays a crucial role in the metabolic reprogramming of cancer cells and the antioxidant response. ROS are usually higher in tumors as a consequence of the accelerated metabolism necessary to grow in limiting conditions. As mentioned, these high levels of ROS can exert both beneficial and detrimental effects, as if they go beyond certain levels, cancer cells become more sensitive to oxidative-stress-induced cell death. Therefore, the activation of the PPP allows them to counteract and balance the levels of ROS through the production of NADPH. Moreover, PPP has shown a context-dependent association in metastatic cells, which need to quickly adapt to metabolic changes to reach their secondary site. In metastatic renal cancers, both the oxidative and the nonoxidative PPP are hyperactivated, while no significant changes have been observed in the progression of breast cancer [31]. Another frequent metabolic alteration which is highly relevant for antioxidant response in cancer is the high demand of the amino acid glutamine, the most abundant nonessential amino acid in plasma (although it becomes essential during pathological stress). Beside its role in protein synthesis, this amino acid represents an alternative source of energy, as it can be used as an anaplerotic substrate for the TCA cycle. Lately, the term ‘glutamine addiction’ has been used to refer to those cancers mostly relying on glutamine as source of energy rather than glucose. Moreover, glutamine is important for the biosynthesis of purines, pyrimidine, asparagine, NAD, and hexosamines, and regulates the cellular redox state. In fact, glutamine can be converted by glutaminase in glutamate, which, remarkably, can be further oxidized to glutathione by glutathione cysteine ligase [32]. Accordingly, it has been shown that the inhibition of glutamine metabolism results in increased ROS, as supported by the evidence of lower levels of reduced glutathione and higher ROS production upon inhibition of glutaminase by bis-2-(5-phenylacetamido-1,3,4-thiadiazol-2-yl)ethyl sulfide (BPTES) [33]. In fact, glutaminase inhibitors have been proposed as a potential anticancer agent for their ability to inhibit the growth of a variety of tumors [34]. The ability of certain cancer cells to sustain survival and proliferation through glucose-independent glutaminolysis, even under hypoxic conditions, is an enlightening example of how metabolic reprogramming can drive tumor progression well beyond the increased glucose metabolism that has been traditionally regarded as a condition enabling tumorigenesis.

## 4. Metabolic Reprogramming, Oxidative Stress and Chemoresistance in Cancer

Mitochondria are essential for each step of tumor progression, from its development to the ability to form metastasis, and eventually, to switch on the mechanisms of chemoresistance [35]. Most cancer cells harbor mutations in the mtDNA, as well as decreased mtDNA content in tumor-initiating cells, i.e., the main drivers of cancer recurrence. These mutations do not completely shut down the mitochondrial energy metabolism and functionality, but they are responsible for the rewiring of common metabolic pathways through a cross-talk between mitochondrial and nucleus signaling [13]. Accordingly, although it is conventionally thought that tumors are mainly glycolytic [4], in recent years, increasing evidence has highlighted the key role played by OXPHOS in a large number of cancers, shedding new light on the metabolic requirements of cancer cells [2,3]. The main concern regarding metabolic rewiring and chemoresistance comes not only from the intertumor, but mostly from the intratumor heterogeneity, with different metabolic signatures coexisting within the same tumor mass, leading to different responses to chemotherapeutic agents [13]. Accordingly, several studies have demonstrated relevant differences in metabolic properties between normal melanocytes and melanoma cells in culture, with the first showing higher sensitivity to chemical inhibitors of mitochondrial respiration, while the second are more sensitive to the inhibition of glycolysis. Furthermore, melanoma cells exhibit a high rate of glutaminolysis. Although most of the melanoma cell lines show a high level of dependence on the classical Warburg effect, there was considerable intertumor heterogeneity, with some cases being more dependent on mitochondrial respiration [36]. Using transcriptomics data, genome-scale metabolic networks, and network topology/controllability analyses, Bidkhori and coworkers were able to stratify HCC patients; this approach allowed them to identify three distinct subtypes with substantial differences in metabolic and signaling pathways, reflecting large differences on the genomic, transcriptomic, and proteomic levels in patient survival [37]. In a similar way, using a panel of high-fidelity, patient-derived xenograft models from hormone-naïve prostate cancer, Lin et al. demonstrated the expression heterogeneity of the genes involved in cellular bioenergetics, and that analogous heterogeneity was also observed in clinical treatment-naïve prostate cancers by analyzing the transcriptome sequencing data [38].

Among others, ovarian cancer attracts attention for its highly heterogenous metabolic scenario. The most aggressive type of this tumor, the high-grade serous ovarian cancer (HGSOC), shows the highest heterogeneity in metabolic phenotype. Gentric and collaborators demonstrated that two main subgroups can be identified in HGSOC, consisting of low and high OXPHOS, with the first one relying on glycolysis and the second on OXPHOS, glutamine, and fatty acid oxidation. This second group shows chronic oxidative stress, which promotes the activation of the transcriptional coactivator PGC-1α. In turn, PGC-1α increases the synthesis of ETC complexes, thereby promoting mitochondrial respiration. Interestingly, this pathway leads to increased response to conventional chemotherapies in the high OXPHOS HGSOCs [39]. On the other hand, we found that cisplatin resistant HGSOC cells lines were more oxidative than their sensitive counterpart, and that inhibition of OXPHOS restored cisplatin sensitivity [40]. We also found that reduced expression of the molecular chaperone TRAP1, known as an important player of metabolic reprogramming of cancer cells, lead to a shift towards OXPHOS, which triggers resistance to cisplatin treatment [40] (see below for a discussion of TRAP1 role in metabolic reprogramming and cancer cell survival). Mechanistically, such a metabolic shift induced the upregulation of the multidrug resistance gene/P-glycoprotein, a well-known membrane protein working as an efflux pump which is able to detoxificate cells from xenobiotics, thus hampering anticancer drug effects [41]. Interestingly, it has been shown that the expression of the ABCB1 gene, encoding for P-glycoprotein, can be induced by the transcription factor Nrf2 [42]. In keeping with the aforementioned data, a recent study on cellular bioenergetic profiling of 13 established and 12 patient-derived ovarian cancer cell lines revealed that chemosensitive cells like the A2780 and PEO1 show a glycolytic phenotype, whereas the resistant ones, C200 and PEO4, showed a switch towards OXPHOS. Accordingly, it was shown that chemoresistant cells were able to survive under conditions of glucose deprivation [43].

However, ovarian cancer is not the only tumor type showing unexpected metabolic features. In fact, it has been reported that melanomas develop alternative metabolic compensatory strategies for survival and growth [44]. Lim et al. demonstrated that targeting a single metabolic component may not be the most effective anticancer therapy, showing that a subset of melanomas rely on OXPHOS and are resistant to oxidative stress through the overexpression of PGC1α. The inhibition of PGC1α resulted in increased ROS production, HIF1α stability, and metabolic rewiring towards glycolysis. Subsequent suppression of glucose metabolism by drug treatments could result in a rewiring in terms of the use of glutamine. The existence of three alternative metabolic strategies ensuring tumor survival highlights the importance of combinatorial therapeutic strategies [44].

Furthermore, HCC has also emerged as one of the tumors relying on OXPHOS. In particular, prolonged exposure of HCC cells to chemotherapeutic agents induces higher level of mitochondrial respiration compared to primary tumors. In fact, using the antiparasitic drug atovaquone, that suppresses the activity of cytochrome bc1 complex, HCC shows decreased growth and survival because of excessive oxidative stress and oxidative DNA damage [45]. Pancreatic cancer also shows crucial dependency on mitochondrial pathways for survival. Viale et al. showed that tumor relapse can occur from a subpopulation of pancreatic tumor cells that, following oncogene ablation, relies on OXPHOS for survival. In particular, they identified a characteristic gene signature in these cells comprising several pathways essential for mitochondrial respiration, as well as a decreased dependence on glycolysis. Accordingly, treatment of resistant cells with OXPHOS inhibitors prevents tumor recurrence [46]. Similar behavior was observed in colorectal cancer. Taking advantage of transcriptomics data of liver metastases, Vellinga et al. observed that cancer cells shift their metabolism towards OXPHOS via the SIRT1–PGC1α signaling pathway in response to therapy, thus suggesting new potential metabolic strategy to prevent disease progression by overcoming drug resistance [47]. Another study demonstrated that the treatment of colon cancer cells with 5-fluorouracil selects for cells that undergo a metabolic reprogramming resulting in addiction to OXPHOS through a reversal of the ratio between pyruvate kinase M2 (PKM2) and PKM1 glycolytic enzymes. This, in turn, correlates with the repression of the pentose phosphate pathway, a decrease in NADPH level, and in antioxidant defenses and the acquisition of stem-like phenotype. Combined treatment with 5-fluorouracil and a pharmacological inhibitor of OXPHOS abolished the spherogenic potential of colon cancer cells and decreased the expression of stem-like markers [48].

Mitochondrial metabolism and consequent ROS generation are also essential to drive anchorage-independent growth, through the regulation of extracellular regulated MAP kinase (ERK)/MAPK signaling pathway, in KRAS-mutated models of lung cancer. Disruption of the mitochondrial function by the loss of mitochondrial transcription factor A reduces tumorigenesis in the same model [49].

As for blood cancers, one example of how metabolic reprogramming affects chemoresistance comes from acute myeloid leukemia. Acute myeloid leukemia cells exposed to chemotherapy adapt their metabolic properties, including different contents of soluble metabolites, higher lipid levels, increased lactate and ATP production, as well as reduced number of mitochondria and lower expression of PGC-1α. These effects are a consequence of VEGFR-2 signaling activation. Accordingly, the inhibition of VEGFR-2 restores mitochondrial biogenesis and function, increasing the response of acute myeloid leukemia cells to chemotherapy [50]. Targeting glutamine uptake also leads to antileukemic responses in acute myeloid leukemia [51]. Primary acute myeloid leukemia cells have an increased mitochondrial mass and oxygen consumption rate compared with normal hematopoietic cells, and targeting mitochondrial translation yields antileukemic effects. The inhibition of mitochondrial respiratory chain complex I by metformin can reduce oxygen consumption and induce cytotoxicity in this disease [52].

The aforementioned studies represent a small number of examples of the high heterogeneity of tumor metabolism. Hence, targeting tumor metabolism might represent a promising strategy, especially when combined with classical chemotherapeutic drugs, to eradicate the cancer cells which are responsible for tumor relapse.

## 5. Metabolic Enzymes and Cancer

As described above, cancer cells show deregulated bioenergetics and an hyperactivation of metabolic pathways such as glutamine metabolism, pentose phosphate pathway, and the TCA cycle, and this involves the activation of oncogenes or loss of tumor suppressors that ultimately contribute to tumor progression and/or chemoresistance. Among others, metabolic enzymes are recently emerging as oncogenes and tumor suppressors themselves, directly or as downstream targets of signaling pathways. One example comes from Hexokinase II (HK2), the first enzyme of glycolysis, that converts glucose to glucose-6-phosphate using cytosolic ATP, which is upregulated in breast cancer. Yang and coworkers demonstrated that this enzyme interacts with and is phosphorylated by PIM2. This phosphorylation enhances glycolysis in MCF-7 cells and contributes to the resistance of breast cancer cells to paclitaxel [53]. Another interesting study showed that sirtuin 5 (SIRT5) promotes glutamine anabolic metabolism and is associated with colorectal cancer cell proliferation, survival, and xenograft tumor growth. In particular, SIRT5 silencing induces apoptosis and cell cycle arrest in colorectal cancer cells, while its overexpression promotes tumor growth in mouse xenograft models. Accordingly, the level of SIRT5 is associated with overall survival of patients with colorectal cancer, suggesting an oncogenic role for this protein. In fact, SIRT5 silencing is able to inhibit the formation of alpha-ketoglutarate from glutamate, which, in turn, inhibits glutamine-derived metabolites from entering the TCA cycle. While being relatively important in normal cells, this process is essential for cancer cells, highlighting a potential tumor-specific metabolic target. Furthermore, the main target of SIRT5 is glutamate dehydrogenase 1 (GLUD1), which is also an important regulator of redox homeostasis; this is consistent with the previously described protective effect of SIRT5 against cellular oxidative stress [54]. Li et al. found that Glutaminase 1 (GLS1), both the known isoforms, kidney-type glutaminase (KGA) and glutaminase C (GAC), is highly expressed in HCC, and that targeting GLS1 reduces the expression of stemness-related genes and cancer stem cells properties in vitro. Even though this might seem inconsistent with the metabolic phenotype of HCC, mostly characterized by low mitochondrial respiration and high glycolytic activity, it is actually due to a vital role played by glutamine in clearing ROS, as well as in satisfying bioenergetic demands and macromolecular synthesis. In fact, low amounts of ROS are needed to maintain quiescence and the self-renewal potential of cancer stem cells; therefore, GLS1 knockout inhibits tumorigenicity in vivo by regulating ROS/Wnt/β-catenin signaling [55]. Cesi et al. demonstrated that inhibition of the RAS/RAF/MEK/ERK pathway in melanoma induces increased ROS levels, which, in turn, activate pyruvate dehydrogenase kinase (PDKs) and the consequent phosphorylation, as well as the inhibition of pyruvate dehydrogenase complex (PDH). This mechanism blocks the further production of ROS by the TCA cycle. Thus, the combination of specific PDK inhibitors with BRAF and MEK inhibitors can help to delay or avoid resistance to BRAF inhibitors by increasing ROS levels and related cell death [56]. In a similar study, Tseng et al. highlighted the role of transketolase (TKT). They found that TKT expression is higher in lymph node metastases, and is associated with poor survival compared with primary tumor or normal tissues of breast cancer patients. Accordingly, TKT depletion, as well as the addition of αKG, enhances the levels of succinate dehydrogenase and fumarate hydratase (FH) which, in turn, lead to a decrease in succinate, fumarate, and HIF1α levels through the stabilization of HIF prolyl hydroxylase 2 (PHD2), thus suppressing breast cancer metastasis. Moreover, the reduction of TKT is responsible for a switch from glycolysis to OXPHOS. Consistent with these results, combinations of the TKT inhibitor oxythiamine, docetaxel, and doxorubicin enhance cell death in triple-negative breast cancer cells [57]. In line with these studies, and consistent with what was anticipated in a previous section in this review, we contributed to shedding further light on the correlation between drug resistance and cancer metabolic remodeling through the characterization in several cancer types of the molecular chaperone TRAP1, which has been extensively described for its role in the stress adaptive response of cancer cells, and has recently emerged as a critical regulator of cancer cell metabolism and chemoresistance.

A summary of the aforementioned metabolic enzymes showing altered expression in cancer and/or involved in anticancer drug resistance is given in Table 1.

## 6. The Mitochondrial Chaperone Tumor Necrosis Factor Receptor-Associated Protein 1

Tumor necrosis factor receptor-associated protein 1 (TRAP1) was initially identified by two independent groups by a yeast-based two hybrid system as a novel protein binding the intracellular domain of the type 1 receptor for tumor necrosis factor and as a molecular chaperone for the retinoblastoma protein [58]. This protein shows a conserved protein architecture with high level of similarity to the heat shock protein 90 (HSP90) [59]. However, despite that, TRAP1 and HSP90 do not share the same functions, and show distinct features. While HSP90 exerts its function mainly in the cytoplasm, TRAP1 is mostly localized in mitochondria, where it contributes to apoptosis protection by several stresses and to the regulation of energy metabolism [60,61]. It is now accepted that this protein also localizes in the cytoplasm, notably being associated with the cytosolic side of the endoplasmic reticulum membrane [62], where it contributes to the control of synthesis of mitochondrial proteins, suggesting a link between its different functions [63]. This ability to affect both cellular biosynthetic and bioenergetic processes makes TRAP1 an interesting protein in the context of tumor biology; however, its role in cancer is controversial. In fact, it was found to be strongly expressed in tumor cells of several adenocarcinomas, whereas normal matched epithelia contain very low levels of this chaperone [64,65,66]; conversely, recent data show a more complex scenario with a lower expression of TRAP1 in lung cancer and cisplatin-resistant ovarian cancer cells (see previous section and [40]).

The TRAP1 has been initially characterized for its capacity to counteract oxidative stress and to protect cancer cells from ROS-mediated damage [67,68]. Indeed, TRAP1 silencing causes ROS accumulation, whereas TRAP1 overexpression decreases ROS production with a consequent increase in reduced glutathione levels [67,68]. In line with this evidence, cells expressing high levels of TRAP1 show increased levels of the scavenging tripeptide GSH, and are more resistant to oxidative stress, also showing cross-resistance to chemotherapeutics [65]. TRAP1 also prevents oxidative-stress-induced apoptosis in neurons as a downstream effector of PINK1; remarkably, the dysregulation of this mitochondrial pathway seems to be involved in the pathogenesis of Parkinson’s disease [69].

Protection from oxidative damage is not the only mechanism by which TRAP1 contributes to cell death protection. In the mitochondria of tumor cells, TRAP1 is involved in a chaperone network that implies the formation of a ternary complex, together with HSP90 and the immunophilin cyclophilin D, that regulates the permeability transition pore opening, maintaining mitochondrial homeostasis, and antagonizing the proapoptotic function of cyclophilin D in permeability transition [70]. In lung cancer, TRAP1 silencing sensitizes cells to mitochondrial mediated cell death driven by VP-16 and β-hydroxyisovalerylshikonin. The same compounds are able per se to reduce TRAP1 protein levels [71,72]. Symmetrically, the overexpression of TRAP1 correlates with resistance to 5-fluorouracil, oxaliplatin, and irinotecan in colorectal carcinoma cells, while treatment with shepherdin, a HSP90 inhibitor which is able to accumulate in mitochondria, reestablishes sensitivity to the same drugs [65]. Accordingly, 5-fluorouracil-, oxaliplatin-, and irinotecan- resistant clones of HT-29 colon cancer cells show increased expression of TRAP1 compared to their sensitive counterpart [65]. As an additional mechanism, the mitochondrial TRAP1 also forms a cytoprotective complex with the mitochondrial isoform of the calcium-binding, antiapoptotic protein Sorcin in colorectal cancer cells [73]. In such a context, TRAP1 is upregulated in drug-resistant colon cancer cells and in paclitaxel- and doxorubicin-resistant breast carcinoma cells [59]. TRAP1 interference in different cancer cell lines (e.g. colorectal and breast cancers), as well as the use of TRAP1dominant negative mutants, sensitizes cells to multiple cell death inducers, ranging from inhibitors of protein synthesis and degradation to endoplasmic reticulum stress inducers and chemotherapeutics agents [59,62]. On the other hand, overexpression of TRAP1 significantly lowers the cytotoxic activity of the mitochondrial complex I inhibitor BAY in BRAF V600E melanoma cells [59]. However, it has been recently demonstrated that TRAP1 expression positively correlates with better response to cisplatin-based therapy in HGSOC, thus suggesting that the effect of TRAP1 on cell viability under pathological condition can depend, at least in part, on its multiple regulatory roles. Among them, particularly interesting is the regulation of cell metabolism which is controversial and, apparently, tumor specific [74].

### Tumor Necrosis Factor receptor-Associated Protein 1 as Driver of Metabolic Rewiring

In the past few years, TRAP1 has also emerged as a critical regulator of mitochondrial respiration through the direct binding to respiratory complexes [74]. Several attempts have sought to elucidate the mechanisms involved in TRAP1 regulation of cell metabolism and, despite the controversy among the different results, it is clear that the role of this protein is highly context-specific [74]. Sciacovelli et al. were among the first to describe a direct implication of TRAP1 in the regulation of cell metabolism. Their results show that TRAP1 behaves as an oncogene, promoting neoplastic progression by binding both complex II and IV of the ETC and reducing complex II activity, the Succinate dehydrogenase (SDH). This in vitro observation was supported by reduced activity of the complex II in colon cancer samples with high expression of TRAP1 as compared to those with low TRAP1. These data candidate TRAP1 as a pro-Warburg factor. Accordingly, TRAP1 silencing increases oxygen consumption rate in osteosarcoma cells, while its overexpression in nontransformed fibroblasts reduces mitochondrial respiration and mimics the respiratory pattern of cancer cells. In particular, the respiratory chain downregulation elicited by TRAP1 interaction with SDH promotes tumorigenesis by priming the succinate-dependent stabilization of the proneoplastic transcription factor HIF1α, independently from hypoxic conditions [75]. Interestingly, a recent study identified S-nitrosylation of Cys(501) in TRAP1 as a posttranslational modification mediating its degradation and activation of SDH, as a consequence of the lack of inhibition by TRAP1 [76]. Protein S-nitrosylation is involved in human HCC pathogenesis; therefore, cells lacking the specific denitrosylase are highly sensitive to SDH inhibitors, candidating SDH as a novel therapeutic target to treat this cancer [76].

A study by Yoshida et al. also supported the view of TRAP1 as a suppressor of mitochondrial respiration, but suggested that this can lead to tumor suppression in specific contexts. Their work shows that TRAP1 reduces oxygen consumption and ATP production and, indeed, its knockout in mouse adult fibroblasts increases oxygen consumption rate and decreases glycolysis. In line with these phenotypes, metabolomic analyses show a marked reduction in the levels of several glycolytic metabolites, while the levels of several TCA cycle metabolites as well as fatty acid oxidation pathway and the NAD+/NADH ratio are increased, consistent with increased oxygen consumption, and suggesting increased metabolic flux through the TCA cycle of TRAP1 silenced cells. TRAP1 silencing also leads to increased activity of the complex IV and ATP production; these results were confirmed in tumor cell lines as HeLa and HCT116. As a consequence, the cells expressing low levels of TRAP1 are exposed to constitutive oxidative stress due to elevated production of ROS by the ETC. These regulations are, at least in part, due to an indirect control mechanism by TRAP1 through the binding to the mitochondrial proto-oncogene tyrosine-protein kinase c-Src, stabilizing its inactive form. Accordingly, c-Src overexpression alone increases the oxygen consumption rate, but the overexpression of both c-SRC and TRAP1 partially reverts this effect [77]. In the same study, the authors reported that a specific set of cancers (renal, bladder, and cervical) shows decreased TRAP1 expression compared to normal counterparts, as opposed to a large body of previous literature, which interpreted these results based on the hypothesis that high oxidative states can contribute to tumor progression and invasion at advanced stages of the disease [77]. Interestingly, and in line with this evidence, we have reported that in a large dataset of HGSOC patients, the expression of TRAP1 is inversely correlated to tumor stage and grade, but directly correlated to the overall survival. Consistent with these data, HGSOC cell lines with clinically-acquired cisplatin resistance show lower expression of TRAP1 compared to their matched sensitive counterpart, derived from the same patient before chemotherapy. Accordingly, high-grade tumors show a metabolic profile which is consistent with mitochondrial metabolism. As a result, TRAP1 silencing induces resistance to platinum treatment, and pharmacological inhibition of mitochondrial metabolism by metformin/oligomycin reverts platinum resistance, providing a causal nexus between TRAP1 expression, oxidative metabolism, and platinum resistance. Mechanistically, TRAP1 silencing leads to the production of proinflammatory molecules, like IL-6 and CSF2 that, in turn, induce increased expression of the multidrug resistance family member TAP1, whose silencing in chemoresistant cells is sufficient to rescue, at least in part, the sensitivity to cisplatin. High expression of TRAP1 in HGSOC cells and specimens also correlates with low expression of markers of epithelial-mesenchymal transition [3].

Very recently, it has been shown that SIRT3 and TRAP1 are overexpressed in glioma stem cells, and that the interplay between them increases activity of the ETC without increasing the ROS concentration in glioma stem cells mitochondria. Inhibition of either TRAP1 or SIRT3 in glioma stem cells leads to metabolic dysregulation, overproduction of ROS, loss of stemness properties, and cell death both in vitro and in vivo. Since active scavenging of ROS is associated with glioma stem cell resistance to radiation and chemotherapy, the interplay between TRAP1 and SIRT3 may contribute to the therapeutic resistance of cancer stem cells in vivo. Moreover, under limited glucose conditions, cancer stem cells switch their metabolism toward the more efficient energy producing respiration to allow adaptation to occur to the harsh microenvironment via a positive feedback loop between TRAP1 and SIRT3; this metabolic characteristic may contribute to the development of therapeutic resistance [78].

On the other hand, Chae et al. reported that TRAP1 is essential for the stability of SDH, and thus, key for the compensatory switch toward oxidative phosphorylation when glucose is limited [79]. Accordingly, it was very recently demonstrated that TRAP1 is transcriptionally controlled by two oncogenic Myc proteins, and that this regulation, in turn, preserves complex II and complex IV folding and function [80]. To recapitulate the puzzling role of TRAP1 in the choice of metabolic signature by tumor cells, Lisanti et al. generated and deeply characterized TRAP1^-/-^ mice. These mice showed an overall reduced incidence of several age-associated pathologies, as well as no tumors compared to WT littermates [81]. Conversely, and in contrast with previous findings, TRAP1^-/-^ mice did not show changes, either in complex I or complex II activity, while both complex III and IV showed higher activity, resulting in increased oxygen consumption levels, ATP production, and a deregulation of mitochondrial respiration. Accordingly, a transcriptome analysis of these mice showed an overall increase in almost all the subunit of the ETC [81]. Surprisingly, TRAP1^-/-^ mice also showed an overall increase in the proteins involved in all glycolysis-related pathways, as well as higher glucose consumption and lactate levels, with a consequent switch to glycolytic metabolism. Despite this, TRAP1 silenced cells are as sensitive as the WT counterpart to apoptosis induced by glucose deprivation and by the glucose analog, 2-deoxyglucose (2-DG). Based on these results, the authors conclude that TRAP1^-/-^ mice encounter a “pseudo-Warburg” phenotype accompanied by a paradoxical increase in mitochondrial respiration, induced as compensation for TRAP1 loss, to ensure the maintenance of mitochondrial homeostasis and bioenergetics in response to unfavorable conditions [81]. In line with this evidence, Yu et al. have recently described a computational systems biology-approach based on an extensive literature search, as well as The Cancer Genome Atlas and single-cell transcriptomics data analysis of multiple cancer types, to model the interconnection between OXPHOS and glycolysis. In brief, their results suggest that the simplest connection between these metabolic states can be represented by the levels of AMPK, HIF1α, and ROS, which lead tumor cells to face three different metabolic states: a Warburg state, characterized by high levels of pAMPK and low levels of HIF1α; an oxidative state, characterized by low levels of pAMPK and high levels of HIF1α; and an hybrid state, characterized by high levels of both pAMPK and HIF1α, ensured by high levels of ROS that stabilizes HIF1α and activates AMPK [82]. The existence of this hybrid state provides tumor cells with an increased plasticity, allowing them to utilize both glycolysis and mitochondrial OXPHOS according to their needs, thus showing a survival advantage compared to cells with the other phenotypes [82]. Moreover, several studies have shown that this hybrid state is able to promote metastasis and the appearance of chemoresistant cancer stem cells [82].

According to these studies, TRAP1 functions in the regulation of ROS levels and cell metabolism, together with its multiple localizations, suggest that this protein might play a key role in metabolic rewiring of tumor cells, acting as a molecular hub for metabolism-related signaling pathways. Indeed, taken together, these puzzling data suggest that TRAP1 is able to either suppress or promote mitochondrial respiration in response to different (oncogenic) signaling pathways, highlighting once more the complexity of the metabolic remodeling that takes place during transformation and tumor progression, and candidating this protein as a crucial player on the stage of responses to metabolic stresses.

## 7. Mitochondria-Directed Therapeutic Strategies

Given their central role in the regulation of cell metabolism, mitochondria have emerged as key therapeutic targets to overcome chemoresistance and selectively kill cancer cells. Accordingly, several compounds have been developed for the targeting of crucial pathways that regulate mitochondrial biogenesis, mitochondrial dynamics, mitochondrial autophagy, and the induction of antioxidant mechanisms. We will focus below on natural compounds and on drugs targeting the specific mitochondrial features of cancer cells that are schematically represented in Figure 1.

### 7.1. Natural Compounds

Numerous natural compounds have shown promising anticancer activity by specifically targeting mitochondrial dysfunction; they work either directly on various components of mitochondria, thus affecting its OXPHOS and apoptotic signaling, or indirectly, by a modulation of metabolic abnormalities which occur upon mitochondrial dysfunction [83]. Curcumin is an active ingredient of turmeric and curry powder, with anti-inflammatory, antiviral, antibacterial, antifungal, and antioxidant properties. Recently, Zhang et al. demonstrated that curcumin induces apoptotic cell death of RN5 murine cells via the AIF-dependent pathway and suppression of the PI3K-Akt-mTOR signaling pathway. In addition, it may suppress tumor growth and angiogenesis without systemic toxicity in vivo, suggesting that it may be a promising antimesothelioma agent [84]. Furthermore, the combination of curcumin with gallic acid, another phenolic natural compound, significantly inhibits cell growth and induces apoptosis in human breast cancer MDA-MB-231 cells, as compared to the dose-effect each drug alone. Treatment with both compounds upregulates the expression of Bax and caspase 3 genes and downregulates the expression of Bcl-2 gene. Moreover, the combination of gallic acid and curcumin decreases GSH levels and increases ROS levels, leading to apoptosis initiation [85]. Sulforaphane, an isothiocyanates molecule, inhibits proliferation of human gastric cancer cells through a complex interplay between cellular mechanisms including regulation of the AMPK pathway, activation of the mitochondrial apoptotic signaling pathway with a decrease in mitochondrial membrane potential and the nuclear translocation of cytochrome c, and the generation of intracellular ROS [86]. Recently, Liang and coworkers demonstrated that shikonin, a naphthoquinone extracted from Chinese herbal medicine plant, induces the generation of intracellular ROS and reduces the mitochondrial membrane potential, ultimately inducing mitochondria-mediated apoptosis in gastric cancer. They also found that the GSH/GSSG ratio was significantly decreased following treatment with shikonin, and that shikonin activates caspase-dependent and -independent apoptosis in gastric cancer cells. Furthermore, shikonin suppresses tumor growth in xenograft animal models and increases lipid peroxidation and apoptosis in vivo. Finally, shikonin enhances the antitumor activities of 5-fluorouracil and oxaliplatin in vivo [87]. Another natural compound of some fruits and vegetables with antitumoral properties in lung cancer, leukemia, and breast cancer is quercetin. Zhou and colleagues demonstrated that quercetin induced apoptosis in the SKOV-3 and A2780 human ovarian carcinoma cell lines through the upregulation miR-145, which, in turn, significantly activated caspase-8 and -9, and cleaved caspase-3. Moreover, this compound inhibited the growth of OVCAR-3 human ovarian cancer cells, which are associated with the expression of VEGF [88].

### 7.2. Drugs Targeting Metabolic Pathways

A large number of drugs, as well as natural compounds, possess antitumoral activity by the targeting of several mitochondria related pathways, such as: i) glycolysis in cancer cells with mitochondrial dysfunction; ii) glutamine metabolism; iii) the mitochondrial retrograde signaling and the ETC; and iv) mutant mtDNA [13]. Among them, drugs targeting glycolysis or the ETC complexes show a very strong effect, especially when used in combination, considering that tumor cells can switch between glycolysis and OXPHOS to support their bioenergetic and biosynthetic needs and, ultimately, to survive during and after anticancer treatments [13]. Indeed, it was recently demonstrated that a combination of two nonsteroidal anti-inflammatory drugs, diclofenac and lumiracoxib, is able to sensitize melanoma cells to the RAF-inhibitor vemurafenib. Both diclofenac and lumiracoxib increase the antiglycolytic impact of vemurafenib and prevent RAF-inhibitor induced metabolic reprogramming towards OXPHOS [89]. A similar study demonstrated that the treatment of breast cancer cells with the Blc2 inhibitors Venetoclax and WEHI-539 leads to decreased OXPHOS in the absence of cell death, whereas it becomes lethal when this treatment is combined with the glycolysis inhibitor 2-Deoxy-D-glucose (2DG), even when cells are grown in three-dimensional spheroids [90]. Chaube and coworkers showed that the inhibition of complex I by metformin in melanoma cells leads to growth arrest because of enhanced glycolysis and consequent extracellular acidification in vitro, but promotes melanoma growth in mice by increasing lactate and VEGF levels. The block of lactate generation in metformin-treated melanoma cells decreases cell proliferation and tumor progression in mice, but the concomitant inhibition of LDH and complex I causes massive apoptosis, increasing synthetic lethality in melanoma [91]. Another relevant function by Metformin is its antitumor effect; in fact, reduced tumor angiogenesis is induced in mice upon metformin treatment [92], which candidate this drug as a potential cancer-preventive agent. Indeed, metformin inhibits the formation of capillary-like networks by endothelial cells in an AMPK-dependent manner [93], although some conflicting data showing upregulation of VEGF in breast cancer and melanoma have emerged, as well as complex effects on the tumor microenvironment [94].

Recently, a phase I study was performed in patients with newly-diagnosed metastatic pancreatic adenocarcinoma to test the PDH inhibitor CPI-613 in combination with modified FOLFIRINOX, with promising results [95]. Similarly, the inhibitor of mutant IDH2 AG-221, and the IDH pan-inhibitor AG881 are currently undergoing clinical development in phase I trials for the treatment of acute myeloid leukemia [96].

With regard to metabolic reprogramming, several studies have underlined the crucial role played by glutamine in cancer metabolism, so that many cancer cells cannot survive in the absence of exogenous glutamine (“glutamine addiction”) [97]. In support of this, several compounds have entered clinical trials to inhibit different steps of glutamine metabolism [98]. The compound CB-839 is an inhibitor of GLS, the enzyme responsible for the conversion of glutamine to glutamate, and is currently being tested in both hematologic and solid tumors, while EGCG and R-162, which inhibit GLUD, the enzyme which catalyzes the conversion from glutamate to alpha-ketoglutarate, that can, in turn, enter the TCA cycle, have been used in preclinical studies of colorectal cancer [98].

### 7.3. Inhibition of Mitochondrial HSP90s as a Therapeutic Strategy

Due to their cytoprotective properties and overexpression in cancer cells, HSP90 chaperones have long been considered attractive targets for anticancer therapy. However, the early compounds developed to inhibit the HSP90 ATPase activity showed an unfavorable toxicity profile, as in the case of Geldanamycin, or poor anticancer activity, as in the case of its nontoxic modified form, 17-AAG (17-N-allylamino-17-demethoxygeldanamycin) [99]. One possible resistance mechanism is the observed compensatory upregulation of HSP70 in response to HSP90 inhibition [100]. To overcome this risk, HSP90 inhibitors have been modified to enter mitochondria, where these compounds cannot induce the upregulation of HSP70 [97]. The first of these compounds was Shepherdin, a peptidomimetic which was initially designed to disrupt HSP90-Survivin interaction, and was later modified to penetrate through the mitochondrial membranes by the addition of an N-terminal helix III homeodomain from the Antennapedia sequence [101]. In tumor mitochondria, Shepherdin interacts with TRAP1 and HSP90, inducing Cyclophilin D-mediated apoptosis in a broad range of tumor cells and multiple xenograft models [102].

A second generation of mitochondria-directed HSP90 inhibitors is represented by Gamitrinibs, a class of fully-synthetic small molecules derived from 17-AAG and modified with different mitochondria-targeting sequences [100]. They cause rapid depolarization, swelling, and finally a disruption of the mitochondrial membrane in a CsA- and CyD-dependent and Bcl2 family-independent manner [100]. In contrast, nonapoptotic concentrations of Gamitrinibs triggers a mitochondrial unfolded protein response, as demonstrated in LN229 glioblastoma cells, where unfolded proteins are found to be accumulated in mitochondria upon treatment with low doses of Gamitrinibs, mirrored by the downregulation of superoxide-dismutase, a marker of mitochondrial stress [103]. This elicits a specific transcriptional program with an increase in several stress-response transcription factors, in particular CHOP, in glioblastoma cells, finally leading to endoplasmic reticulum stress response and autophagy [103]. It is worthy of note that Gamitrinibs treatment significantly reduced energy consumption of LN229 glioblastoma cells and reduced glucose consumption and lactate production, also compromising the tethering of the glycolytic enzyme HK2 to the mitochondrion, which is required for glycolysis and for coupling glucose metabolism to oxidative phosphorylation [104].

Taken together, the above data suggest that the targeting of cancer cell bioenergetics holds great potential for therapeutic interventions. Based on such accumulating evidence, a significant effort has been made in recent years to find other efficient inhibitors of mitochondrial HSP90 functions. Several new derivatives were evidenced to accumulate in the mitochondrial fraction of colon carcinoma cells, and one compound in particular was proven to be good at inhibiting recombinant TRAP1 ATPase activity and at reducing cell growth and inducing apoptotic cell death in colon carcinoma cells [105]. On the other hand, the report of the crystal structure of TRAP1 has opened new scenarios regarding the possibility to selectively target this specific HSP90 paralog in cancer cells. This could also make it possible to inhibit cytosolic TRAP1 functions without affecting HSP90 activity and, possibly, preventing compensatory HSP70 upregulation. Finally, a new class of compounds that, in contrast, allosterically stimulate HSP90/TRAP1 ATPase activity has been recently synthesized, showing anticancer efficacy in the low micromolar to nanomolar range [106]. Of note, treatment with these compounds decreases SDH activity [103], suggesting that they should be tested in tumors in which low levels of TRAP1 have been correlated with an unfavorable prognosis.

Some of the most effective compounds targeting metabolic pathways in mitochondria are shown in Figure 1.

## 8. Concluding Remarks

Several cancer cell types may undergo different bioenergetic changes, some to more glycolytic and some to more oxidative. In this review, we focused on different metabolic signatures in the context of cancer chemoresistance, paying special attention to the role of TRAP1 chaperone in cancer metabolic regulation. Chemoresistance represents the main obstacle to the cure of oncological patients. Today, increasing evidence suggests the need of extensive reprogramming of mitochondrial activity in the development of cancer drug resistance. Therefore, therapeutic strategies targeting different metabolic events specific to cancer cells, including glycolysis, glutaminolysis, OXPHOS, are strong potential candidates to prevent chemoresistance. ROS have also been implicated in the metabolic reprogramming of cancer cells, allowing adaptation to occur to oxidative stress that ultimately promotes tumorigenesis and chemoresistance. Indeed, redox homeostasis and the antioxidant defense of cancer cells, as well as their apoptotic response to oxidative stress-inducers and DNA-damaging anticancer drugs, are considered other important processes in cancer. These studies may lead to approaches aimed at counteracting adaptive responses of mitochondria following chemotherapy, thus contributing to increased efficacy of anticancer strategies. Finally, this review, besides summarizing how metabolism, oxidative stress adaptation, and chemoresistance are strictly interconnected, also shows how TRAP1 stays at the crossroads of these processes, thus shedding new light on the molecular networks which form the bases of several cancers. The apparent contradictions that still make TRAP1 an attractive cancer biomarker will be unveiled after deep analyses of the TRAP1 pathway in individual cancer types and upon careful characterization of unique/selective TRAP1 client proteins in the regulation of cancer cell metabolism and drug resistance.

## Figures and Tables

**Figure 1 biomolecules-10-00135-f001:**
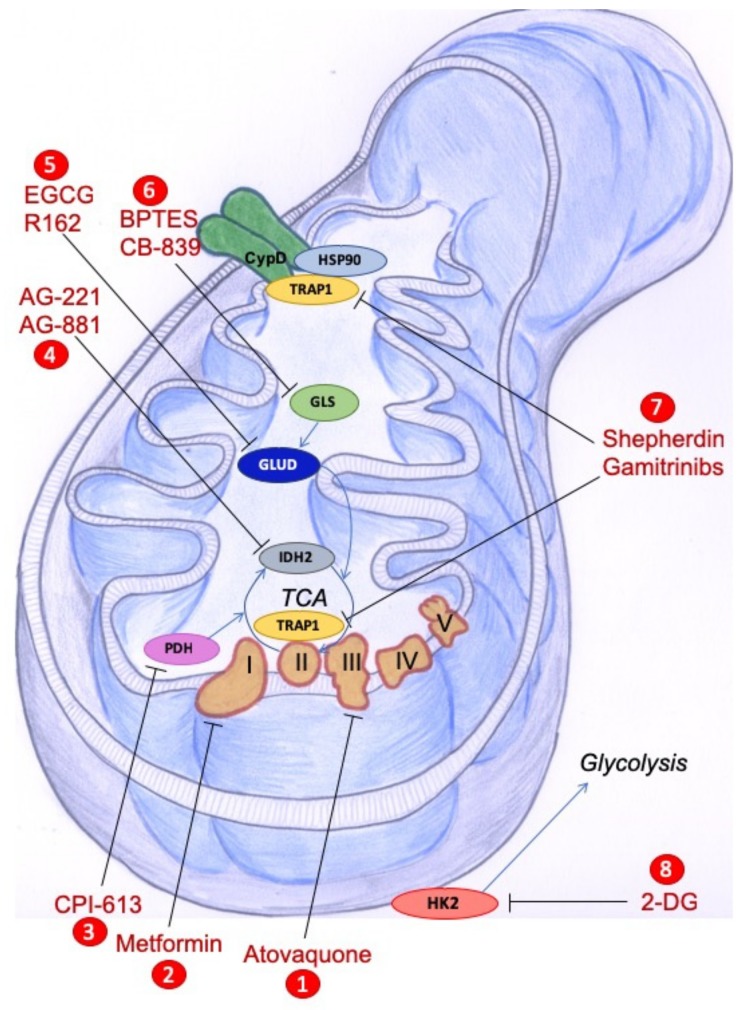
Schematic representation of agents targeting components of mitochondrial metabolism that have been proposed in cancer treatment (either approved, already in clinical trials, or tested in preclinical models). Only the mitochondrial targets addressed in this review are indicated in the figure. Specifically, (1) atovaquone is a FDA-approved antimalarian drug, whose repurposing proposed it as mitochondrial complex III inhibitor in cancer cells; (2) metformin inhibits mitochondrial complex I; (3) CPI-613 targets catalytic and regulatory functions of the PDH complex, the enzyme converting pyruvate into acetyl-CoA to be used in the TCA cycle; (4) AG-221/AG-881 compounds target mutant IDH2; (5) EGCG/R162 block TA/GLUD, that converts glutamate to the TCA cycle intermediate alpha-ketoglutarate or (6) BPTES/CB-839 inhibition of GLS that converts glutamine to glutamate, have shown efficacy in preclinical studies; (7) shepherdin and gamitrinibs are inhibitors of the mitochondrial chaperones TRAP1 and HSP90, that play roles in ETC complex stability and activity and in resistance to apoptosis through the binding and stabilization of CypD; (8) 2-DG inhibits glycolysis by competing with glucose as substrate for HK2, a mitochondrial membrane associated enzyme in cancer cells. CypD: Cyclophilin D; GLS: Glutaminase; GLUD: Glutamate dehydrogenase; HSP90: Heat Shock Protein 90; IDH2: Isocitrate dehydrogenase 2; PDH: Pyruvate dehydrogenase; TCA: Tricarboxylic acid; TRAP1: TNF Receptor-Associated Protein 1.

**Table 1 biomolecules-10-00135-t001:** Metabolic enzymes involved in oxidative stress and energy metabolism whose increased expression correlates with neoplastic transformation, prognosis and/or chemoresistance in different tumors. BC: Breast Cancer; CRC: Colorectal Carcinoma; DLBC: Diffuse Large B cell Lymphoma; EC: Endometrial cancer; HCC: Hepatocellular Carcinoma; LC: Lung Cancer; MESO: Mesothelioma; OC: Ovarian Cancer; OS: Osteosarcoma; PC: Prostate cancer.

Protein	Function	Tumor Type	References
Thioredoxin-1 (Trx1)	Redox protein member of thioredoxin system which plays a crucial role in the cellular redox homeostasis	DLBCL	[24]
		CRC	[107]
Thioredoxin reductase-2 (TrxR2)	Member of the family of pyridine nucleotide-disulfide oxidoreductases, component of the antioxidant thioredoxin system	HCC	[108]
		OS	[109]
		LC	[110]
Peroxiredoxin 3 (Prx3)	Antioxidant enzyme with mitochondrial localization, member of the thioredoxin system	BC	[111]
		HCC	[112]
		MESO	[113]
		EC	[114]
		PC	[115]
SIRT1	NAD^+^-dependent protein deacetylase, plays key roles in DNA damage response and metabolic adaptation to energy stress. Along with PGC1α, is involved in chemotherapy-induced shift to OXPHOS in CRC cells	CRC	[47,116]
SIRT5	NAD-dependent protein lysine demalonylase, desuccinylase and deglutarylase able to remove malonyl, succinyl, and glutaryl groups from the lysine residues of proteins	CRC	[54]
Hexokinase II (HK2)	Catalyzes the phosphorylation of glucose to generate glucose-6-phosphate in the first step of glycolysis. It promotes chemoresistance by enhancing cisplatin-induced autophagy	OC	[117,118]
Transketolase (TKT)	Ezyme catalyzing important reaction both in the Calvin cycle and in pentose phosphate pathway	BC	[57]

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
