# Peer review of "Modulation of Mitochondrial Metabolic Reprogramming and Oxidative Stress to Overcome Chemoresistance in Cancer"

_biomolecules, 2020, doi:10.3390/biom10010135_

Round 1

Reviewer 1 Report

Submitted paper by Avolio et al. discussed factors underlying chemoresistance as a major cause of cancer recurrence and metastasis. They discussed many factors such as mitochondrial factors, oxidative stress, enzymatic factors.

In general the article is nicely written and clear for the reader. I suggest adding something about P-glycoprotein and its role in chemoresistance.  

Authors talked about the anticancer effect of metformin in experimental studies. I suggest to add something about its antiangiogensis effect for examples: Zafaar et al., 2014 published this articel in Plos one. 

https://journals.plos.org/plosone/article?id=10.1371/journal.pone.0100562 

and some similar articles are there. 

Number of abbreviations are huge, I would like to ask authors to reduce the number of abbreviations to those are used extensively in the article.

Reviewer 2 Report

This is a clearly written and comprehensive review of the literature on the role of metabolic re-programming and oxidative stress in the development of chemoresistance in cancer, with particular focus on the involvement of the molecular chaperone TRAP1. The authors provide a very thorough background of the basic concepts of mitochondrial structure and function and their role in ROS production, which would be useful to non-specialists. The key mechanisms of metabolic adaptation and maladaptation are discussed with examples of their role in chemoresistance in different cancers. The current evidence on the dichotomous role of TRAP1 on the regulation of mitochondrial metabolism is presented and possible interventions to overcome chemoresistance, targeting metabolic reprogramming and TRAP1, are proposed.

My main criticism is that the authors have recently published two other reviews covering the role of TRAP1 in metabolic reprogramming in cancer and its role in drug resistance (Matassa et al, Genes (Basel). 2018 Apr; 9(4): 195, and Amoroso et al, Advances in Protein Chemistry and Structural Biology, Volume 108, 2017, Pages 163-198).

 I have the following comments/suggestions for improving the manuscript.

I believe that Section 3 on mitochondrial dysfunction and diseases is not needed and it should be removed, as it is not directly relevant to the subject. The authors should instead include a section introducing the key cellular antioxidant mechanisms and discussing how metabolic re-programming influences antioxidant responses. This could include the role of glutamine and the pentose phosphate pathway in glutathione biosynthesis and metabolism.

The authors should discuss glutathione as a key regulator of cellular oxidative stress and drug resistance and its possible role as a therapeutic target.

On page 5, line 227 the sentence should read: “Among the others, ovarian cancer attracts….”

In table 1, page 8 the sentence should read: “Along with PGC-1α, is involved in chemotherapy-induced shift…”

On page 13, line 589 the authors should define the acronym 17-AAG.

The statement: “…., also detaching the glycolytic enzyme HK2, where it needs to be located in cancer cells to exert its function” on page 14, lines 599-600, is unclear and should be re-written.

On page 14, line 601 the sentence should read: “Altogether, the above data suggest that targeting of cancer cell bioenergetics….”.

On page 15, line 622 the sentence should read: “metformin inhibits mitochondrial complex I”.

Round 2

Reviewer 1 Report

Authors adequately followed the Reviewer's comments